# Assessing Park Quality with a Wearable Video Device and an Unmanned Aerial System

**DOI:** 10.3390/ijerph191811717

**Published:** 2022-09-16

**Authors:** Richard R. Suminski, Gregory M. Dominick, Eric Plautz

**Affiliations:** Department of Behavioral Health and Nutrition, University of Delaware, Newark, DE 19726, USA

**Keywords:** observation method, health behavior, measurement, parks

## Abstract

Parks are ideal places for promoting physical activity, which is vital for achieving and sustaining good health. Thus, it is important to develop and provide the best methods for assessing aspects of parks that could influence physical activity. This study examined the use of high-tech video capture for describing park quality. Videos were obtained with a wearable video device (WVD) and an unmanned aerial system (UAS) at 28 and 17 parks, respectively. In-person audits of park attributes were performed using the Physical Activity Readiness Assessment (PARA) instrument while video was simultaneously captured. The PARA provides quality ratings of park attributes that range from poor to good. Kappa statistics were calculated to compare in-person PARA outcomes with PARA outcomes obtained by reviewing the WVD and UAS videos. Substantial and almost-perfect agreements were found between WVD and in-person PARAs on the quality of features and amenities and the severity of incivilities. Agreements between UAS and in-person PARAs on feature and amenity quality and incivility severity were unacceptable (mostly fair and moderate). In conclusion, being able to reliably assess park quality using video provides advantages over in-person assessments (e.g., retrospective analysis). In addition, it sets up the possibility of utilizing computer vision to automate the video analysis process.

## 1. Introduction

Regular physical activity (PA) can prevent serious health problems such as obesity, cardiovascular disease, diabetes, and some cancers, and promote muscular and cardiovascular fitness, mental health, and psychological wellbeing in individuals across the lifespan [1,2,3,4]. Despite this strong evidence, a substantial proportion of children, adults, and older adults are considered physically inactive. For instance, population-based estimates from accelerometer-derived data indicate that nearly 90% of adults are inactive, with over half of waking hours spent in sedentary behaviors (55% or 7.7 h/day) [5,6]. Similarly, objective PA assessments in older men and women (70+ years) show that the prevalence of achieving 150 min/week of moderate-to-vigorous PA is approximately 12.7% [7]. Children under the age of 19 fare better; however, less than 45% fail to engage in an hour of PA or more per day [3,8]. Increasing the percentage of people across the lifespan who engage in PA is a national priority for disease prevention. 

Community-level interventions are highly recommended and are becoming the “approach of choice” for promoting PA [9,10]. Evidence suggests that targeting aspects of the built environment (i.e., man-made structures and features) enhances the effectiveness of these interventions for supporting PA [9,11,12,13]. Parks are a built environment feature considered ideal for PA. They are often in residential neighborhoods, offer several amenities for PA, and available year-round to residents of all ages, at no or low cost [14,15,16,17]. Given that there are 1,829,533 park acres within the 100 most populous cities in the U.S. and that parks are within walking distance for over 85% of residents of the 10 largest U.S. cities, parks are arguably one of the most important resources available for PA promotion [18]. 

Although access to parks is directly associated with PA, park attributes such as size, number, and type of features (particularly basketball courts, playgrounds, and sports fields), as well as their condition, encourage greater park use, moderate-to-vigorous PA, and energy expenditure [15,19,20,21]. Even nearby parks are less likely to be used if the attributes are outdated or of poor quality [15,20]. Longitudinal studies suggests that renovations of parks are followed by increased park use for PA [7,22,23]. Other park characteristics, such as aesthetics and auxiliary features/amenities (e.g., shelters, water fountains, benches) do not appear to directly impact park-based PA. However, it is plausible that these characteristics moderate the association between feature quality and use, but there is no definitive evidence to support this claim [24]. 

Elucidating how parks can be optimized for promoting PA requires a multifaceted approach, including the accurate measurement of park attributes. Direct observation or auditing is the most widely used approach for this purpose [25,26,27]. The techniques employed vary, but generally require a person to canvas parks while counting park attributes and referring to various criteria to determine their quality. Although reliable and sometimes considered a “gold standard” for measuring park attributes, existing observation methods have well-known limitations that could be addressed with today’s advanced, technological solutions. For instance, training and data collection are burdensome; reliability can be variable in large-scale and longitudinal studies; a person only records what they are asked to record in that moment with their perceptions/beliefs/experiences as filters; and it can be difficult to obtain data on park attributes while recording their use, particularly with large groups [26,27,28,29]. 

The integration of video and wearable technology has led to the creation of highly mobile devices ideal for objectively measuring georeferenced imagery, including park visitors and aspects of park attributes, in real-time. Video recorders have been embedded into eyewear frames (wearable video devices (WVD)) providing point-of-view assessments or attached to aviation devices (unmanned aerial systems (UAS)—drones). With these devices, data captured in public spaces can be obtained in a more unobtrusive, less unnoticeable manner. Recent studies suggest that assessments of human activity in parks and on sidewalks/streets and disorder indicators in neighborhoods (e.g., yard trash) are as or more accurate than in-person methods when WVD or UAS technology is utilized [28,30,31,32,33,34]. Other advantages associated with using video to assess parks are the greater unobtrusiveness of obtaining video (WVD video not UAS video) versus carrying clipboards and writing down information, the ability to conduct retrospective analysis of video data, and the potential automation of video analysis using computer vision (CV) algorithms developed with machine learning techniques such as deep convolutional neural networks [35]. The latter advantage would substantially improve our understanding of park use for PA by increasing data collection and analysis capacities while maximizing data reliability. However, an essential step in developing such algorithms involves the process of annotation, which requires a human to label images in a video frame or picture (e.g., identify/label a park attribute and rate its quality) to train a machine learning model. Prior to this, evidence is needed showing that WVDs and UASs are suitable for capturing images of park attributes that can be identified accurately by a human video reviewer. Therefore, this study determined if video of park attributes obtained with a WVD and a UAS provide the information needed to accurately assess the quality of a park’s features and amenities and the severity of park incivilities. Outcomes from video analyses were compared with outcomes obtained during in-person park evaluations. 

## 2. Materials and Methods 

### 2.1. Data Collection Devices

#### 2.1.1. Wearable Video Device–Gogloo E7 SMART

The Gogloo Engine E7 is a state-of-the-art, point-of-view WVD, indistinguishable from a pair of normal sunglasses (Figure 1) [Gogloo E7 SMART eyewear, model number E7B0100, https://gogloob2c.com/ (accessed on 13 September 2022)]. They have a durable water-resistant frame and a 110-degree 8MP SONY camera in the bridge of the glasses that records high-definition video in 1080p at 30 frames per second or 720p at 60 frames per second. The Gogloo Engine E7 smart glasses have an integrated rechargeable battery that can record up to 8 h and the capacity to support up to a 64 Gb removable micro SD card that allows for the capture of hours of video footage. All recorded video can be readily downloaded and saved as files with time/date/GeoCoordinate stamps. The Gogloo Engine E7 was selected over other commodity WVDs (e.g., GoPro) and stationary cameras because it captures video almost exactly from the individual user’s point of view (camera is between their eyes), and thus theoretically acquires the same information as an in-person auditor 

#### 2.1.2. Unmanned Aerial System (UAS) 

The DJI Mavic 2 Pro is a compact (322 × 242 × 84 mm (length × width × height)) UAS (Figure 2). It has a flight time of approximately 31 min and uses a GPS/GLONASS satellite positioning system. It has a maximum wind-speed resistance of 10.5 m/s with omnidirectional obstacle sensing, allowing for left/right, up/down, and forward/backward obstacle sensing for increased safety while in flight. The integrated Hasselblad camera has a field of view of 77°, 28 mm (35 mm format equivalent), f/2.8–f/11 aperture, and auto focus at 1 m to ∞, and can record up to 4K at 3840 × 2160p with a gimbal that has a 3-axis (pitch, roll, and yaw) stabilization for increased viewing angles and clarity. The UAS also included a controller equipped with OcuSync 2.0 that provides stable connectivity between the controller and the Mavic 2 Pro. The controller supports automatic switching between 2.4 GHz and 5.8 GHz, reducing the influence of environmental interference on UAS operation and image quality. This also ensures reliable long-range transmission at distances of up to 10 km. The controller has a 5.5-inch 1080p screen providing an ultrabright display to keep live feed easily viewable, especially in direct sunlight. Finally, the controller uses a 5000 mAh battery for up to 2.5 h of operation and it supports the DJI GO 4 Fly App.

### 2.2. Parks

Twenty-eight neighborhood parks in New Castle County, Delaware, USA were randomly selected for study from all neighborhood parks in the county (n = 72; M = 9.1 acres, SD = 8.8 acres). All study parks contained features and amenities and ranged in size from 3.7 to 46.9 acres (M = 12.2 acres, SD = 12.6 acres). The focus was on neighborhood parks because they are the most common park type; they serve a considerable proportion of the population as compared to mini parks (<1 acre in size) and larger, regional parks (>50 acres in size), and they contain most of the features (e.g., basketball courts, tennis courts, playgrounds) normally used for PA [36]. Due to Federal Aviation Administration (FAA) prohibition on UAS in airport no-fly zones, the UAS could only be used at 17 of the 28 study parks. The excluded parks, however, were similar to the 17 parks included in terms of acreage and types of features/amenities.

### 2.3. The Physical Activity Resource Assessment (PARA) 

The PARA is a one-page checklist developed to assess 13 park features for PA (e.g., baseball field), 12 park amenities (e.g., benches), and 12 incivilities (e.g., litter) that could reduce the pleasure of using a park [27]. The quality of features and amenities, if present, is rated according to specific operational definitions and a scale ranging from 1 “poor” to 3 “good”. Incivilities are rated for severity according to incivility-specific definitions of severity that range from 0 “not present” to 3 “most severe”. In the current study, the auditory annoyance incivility was not included. Lee et al. [27] reported that the instrument has good inter-rater reliability (*rs* > 0.77), requires minimal time to complete, and provides easily interpretable data.

### 2.4. Procedures

Prior to data collection, two data collectors (authors RRS and EP), who are experts on the use of the PARA, trained to become proficient at operating the UAS and WVD. In addition, both data collectors identified the type and location of each park feature and amenity using Google Maps and site visits. Maps of each park were created showing the areas to be evaluated (e.g., playground with a line drawn around it demarcating the evaluation area) and the order of evaluation (devised to minimize distance between attributes). The 28 parks contained 128 features with playgrounds constituting the majority (43.8%) followed by basketball courts (15.3%), tennis courts (11.8%), baseball fields (9.7%), greenspaces (8.3%), and sidewalks/trails (6.9%). Bike racks, exercise stations, splash parks, and football fields each constituted less than 2%, and there were no soccer fields, pools, volleyball courts, or sandboxes present. There were also 148 amenities. Benches were the most common (21.9%), followed by trash cans (16.9%), access points (8.7%), picnic tables (6.8%), benches (5.5%), and lighting (2.7%). Landscaping efforts, missing grass, and overgrown grass were evaluated at all the parks and only within feature and amenity evaluation areas. Bathrooms, fountains, sandboxes, and showers/locker rooms were not assessed because they were either not present or closed. In the subset of parks where the UAS was flown, there were 75 features and 82 amenities. The types of features and amenities were similar to those found at all 28 parks. 

During the summer months of June and July, the two data collectors visited each study park twice between 10 am and 3 pm. During visit 1, one data collector conducted in-person PARAs while the other operated the WVD. The WVD protocol required the WVD user to continuously record while closely examining features and amenities along with the surrounding, affiliated areas. The WVD user did not write down information nor evaluate the quality of the park attributes. 

A second visit was made by the same two data collectors to the 17 parks where the UAS could be used. During these visits, the in-person PARA was completed by one data collector while the other operated the UAS. Altitude was set at 7.6 m because this was the closest to the ground that the device could be flown safely (e.g., safe distance from feature/amenity tops). Speed did not exceed 4.8 kph. The UAS operator flew the device to each feature and amenity and utilized two flight patterns to verify the presence/absence of PARA criteria without writing down information. Flight patterns consisted of 360° flights around attributes and row flights involving end-to-end attribute flyovers. Both flight patterns were used at all attributes, except sidewalks/trails, where the device was flown only in one direction along their lengths. 

All data collectors carried park maps delineating areas to be evaluated. The video device user began assessments, followed by the in-person auditor, who waited until the device user moved on to evaluate the second attribute. This order was selected because the device user needed less time to complete an evaluation. Data collectors did not communicate with each other during data collection and they did not observe one another conducting evaluations.

After all park assessments were completed, a study team member highly trained and experienced with the PARA—but not involved with data collection—utilized a video-reviewing computer software program and the PARA instrument as a guide to assess the quality/severity of park attributes. All tools available with the software (e.g., rewind, stop, zoom) were used during video assessments. Videos were reviewed by two different study team members, also well-versed in the use of the PARA, after comparisons with in-person PARAs were complete in an attempt to determine the cause of any discrepancies. 

### 2.5. Statistical Analysis

Data distributions for each variable were normally distributed. Weighted kappa statistics (*κ*) were derived to assess agreement between in-person PARA and WVD and UAS PARA quality ratings of features and amenities and severity ratings of incivilities [37]. The in-person PARA was considered the criterion given the novelty of using WVD and UAS videos for assessing park quality. The following value ranges put forth by Landis and Koch [38] were used to evaluate agreement based on the *к* statistics: <0 poor, 0.0 to 0.2 slight, 0.21 to 0.4 fair, 0.41 to 0.6 moderate, 0.61 to 0.8 substantial, and 0.81 to 1.0 almost perfect. All statistical analyses were performed using the SPSS statistical software package with alpha set a priori at 0.05 (IBM Corp. 2015. SPSS Statistics for Windows, Version 23.0. IBM Corp, Armonk, NY, USA).

## 3. Results

### 3.1. Feature Quality: In-Person versus WVD

The κ for the 128 features rated during in-person and WVD PARAs was 0.84 (*p* < 0.001) indicating almost perfect agreement between method. For in-person feature evaluations, 7.0% were rated poor, 29.7% were rated moderate, and 63.3% were rated good; and for WVD PARAs, 6.3% were poor, 31.5% were moderate, and 62.2% were good. For the individual features, substantial agreements on quality ratings between methods were found for baseball fields, tennis courts, and trails, while almost perfect agreement was observed for basketball courts and playgrounds (Table 1). 

### 3.2. Feature Quality: In-Person versus UAS

The κ for the 75 features rated during both in-person and UAS PARAs was 0.48 (*p* < 0.001), which is moderate agreement. Features during in-person PARAs were given poor ratings 10.4% of the time, moderate ratings 29.9% of the time, and good ratings 59.7% of the time; and during UAS PARAs, 5.2% were given poor ratings, 33.8% were given moderate ratings, and 61.0% were given good ratings. Kappas for individual features are in Table 1. Agreements between methods on the quality of trails and playgrounds were fair. Moderate agreements were found for baseball fields, and substantial agreement between methods was observed for tennis and basketball courts.

### 3.3. Amenity Quality: In-Person versus WVD

The κ for the 148 amenities rated during in-person and WVD PARAs was 0.79 (*p* = < 0.001), indicating substantial agreement. During in-person PARAs, 10.6% of amenities were rated poor, 23.8% were rated moderate, and 65.6% were rated good. Similarly, 9.9% of amenities were rated poor during WVD PARAs, 24.5% were rated moderate, and 64.9% were rated good. Agreements on quality ratings of the individual amenities are provided in Table 1. Substantial agreement between methods was noted for benches, picnic tables, and shelters, while almost perfect agreements were found for access points, landscaping efforts, lighting, and trash containers.

### 3.4. Amenity Quality: In-Person versus UAS

The κ for the 82 amenities rated during both in-person and UAS PARAs was 0.40 (*p* < 0.00) or fair agreement. The UAS PARAs found 7.7% of amenities in poor, 22.1% in moderate, and 69.2% in good condition; whereas the in-person PARAs noted 8.2% poor, 27.5% moderate, and 64.3% good condition. Kappas for individual amenities were generally low (Table 1). Agreement on the quality of picnic tables and landscaping efforts was fair; for access points, benches, shelters, and lighting it was moderate; and for trash containers it was substantial.

### 3.5. Incivility Severity: In-Person versus WVD

A total of 308 incivilities ratings (65.1% not present) were made during in-person and WVD PARAs. Overall, the two methods displayed almost perfect agreement on incivility severity (κ = 0.86, *p* < 0.001). Similar percentages of severity levels were noted between methods: in-person—64.6% not present, 21.8% good, 9.7% mediocre, and 3.9% poor; WVD—65.6% not present, 20.1% good, 9.1% mediocre, and 5.2% poor. Agreements for the individual incivilities are shown in Table 1. Moderate agreement was found between methods on dog refuse; substantial agreement was noted for broken glass, alcohol evidence, graffiti, litter, and vandalism; and almost perfect agreement was found for grass presence/absence and overgrown grass. Kappas for unattended dogs, evidence of substance abuse, and evidence of sex paraphernalia could not be calculated due to one or fewer occurrences. However, the absolute agreements for these incivilities were all 100%.

### 3.6. Incivility Severity: In-Person versus UAS

In the parks where the UAS was flown, 187 ratings of incivilities (71.9% not present) were made during both in-person and UAS PARAs. The overall kappa was 0.52, with *p* < 0.001 indicating moderate agreement. The UAS ratings of severity were somewhat dissimilar from in-person ratings, in that 80.2% were categorized as not present during UAS PARAs, 9.6% were rated as good, 6.4% were rated as mediocre, and 3.7% were rated as poor. Conversely, 63.6% were deemed not present during in-person PARAs, 24.1% were rated as good, 10.2% were rated as mediocre, and 1.1% were rated as poor. Kappas for each incivility rated are given in Table 1. Fair agreement was found for overgrown grass, grass presence/absence, and graffiti; and slight agreement was found for litter. The other incivilities were observed one or fewer times. The absolute agreements were: 76.5% broken glass, 88.2% dog refuse, 100% unattended dogs, 64.7% alcohol evidence, 100% substance abuse evidence, 100% sex paraphernalia, and 100% vandalism.

## 4. Discussion

The aim of this study was to determine if the quality of park attributes can be accurately ascertained from videos obtained with a WVD and a UAS. Results indicate that WVD videos are viable for assessing the quality of parks features and amenities and the severity of park incivilities. Conversely, the UAS videos were not adequate for assessing most park attributes.

The quantity and quality of park features and amenities, as well as overall park attractiveness, are associated with its use for PA [7,15,19,20,21,22,23]. Video has been used to augment assessments of PA occurring in parks and on sidewalks/streets, but it has not been employed for park attribute assessment [27,30,32,33]. In the current study, videos captured with a WVD contained information for accurately assessing the quality of park features and amenities as well as the severity of park incivilities. The overall kappa values denoted substantial to almost-perfect agreement between in-person and WVD PARAs. Averaged across features, amenities, and incivilities, the kappa value was 0.79, which is consistent with the high, positive correlation (rs > 0.77) found during PARA development [27]. Retrospective reviews of WVD videos indicated that the few discrepancies between methods on ratings of features and amenities were due to differences in how the individuals performing the PARAs interpreted the quality criteria. For example, out of 17 tennis courts examined, disagreement occurred twice and involved one evaluator rating a court good (court surface and nets are in fairly good condition) and the other rating it mediocre (court surface and nets are in need of some repair, but usable). Given that the PARA criteria are somewhat ambiguous and require users to make “subjective” judgements, a small number of disagreements between evaluators would be expected and considered acceptable from a measurement perspective. For incivilities, overall agreement was substantial, but there were some incivilities where only moderate agreement was noted (broken glass, dog refuse, alcohol evidence, graffiti). A re-review of the videos suggested that when discrepancies occurred, they were the result of a mismatch in counting of the incivilities, and thus in the assignment of severity ratings. For example, when assessing alcohol evidence at one park, the in-person auditor counted two bottle caps (mediocre rating) while the WVD reviewer counted one bottle cap (good rating). Over and undercounting of incivilities occurred at similar rates between methods, suggesting that most of the disagreements on the severity of incivilities were related to inherent problems with the PARA criteria and the requirement to precisely count small details. Therefore, the majority of discrepancies between methods were associated with what could be considered “flaws” in the PARA that affect inter-rater reliability regardless of whether the PARA is conducted in-person or using a WVD video.

The UAS PARA outcomes agreed less with in-person outcomes than the WVD outcomes did. Most quality ratings of attributes using UAS videos demonstrated only fair agreement with ratings from in-person PARAs. These results suggest that UAS videos obtained following our protocol were not adequate for assessing park quality. This is in contrast to others who have successfully used UASs to measure environmental conditions and the presence of humans in specific behavior settings [30,31,32,33,34]. For example, Grubesic et al. [30] deployed a UAS at an altitude of 65 m to accurately detect several different objects on the ground along streets and alleyways. The contrast in findings could be due to several reasons. In the Grubesic et al. [30] study, large objects (e.g., piano) and grouped items (e.g., piles of trash) were mostly identified. Moreover, it appears from their published photos that their evaluation areas were virtually free of grass (e.g., dirt yard) making it easier to discern items, and they did not determine how much view obstructions (e.g., trees) affected their ability to see items with the UAS. In the current study, the PARA required only a few (<4), mostly small (e.g., contraceptive) items to be identified. Further, the items could be scattered around an evaluation area and/or hidden from a UAS due to aerial view obstructions. Indeed, retrospective evaluations of the UAS videos indicated that most discrepancies were due to the difficulty in identifying minute details (six incivilities required small items to be counted) and not being able to capture criteria that were “under” an attribute (e.g., the condition of bench bottoms, inside a shelter roof (where graffiti is prevalent), and undersides of playground equipment). It is possible that some of these problems could be addressed using a more sophisticated UAS (one that has oblique imagery for obtaining videos at an incline or a camera with a greater resolution) and/or developing different video capture protocols. However, other notable disadvantages to using a UAS to evaluate parks may also need to be addressed: the relative cost (in this study the UAS was 10 times the cost of the WVD), flight restrictions (40% of parks in the current study were in no-fly zones), the training required (some U.S. states may require a pilot license), and local ordinances limiting UAS use.

As mentioned earlier, there are advantages to enhancing observation and audit methods with video. First, when using a WVD, the data collector is less conspicuous because the device closely resembles a pair of everyday eye/sunglasses. In addition, the data collector is not overtly displaying signs of collecting data such as carrying a clipboard or writing down information, etc. Second, the WVD is also very easy to use and our data collector needed only about one minute to learn all of its functions. Third, the WVD videos represent a theoretically permanent record of the park conditions at the time that the video was obtained. When an in-person audit is performed, the information used to determine if quality criteria were met is lost. What the data collector sees during a park visit is condensed into PARA quality categories with limited descriptions and there is no record of the actual park attribute conditions. Videos, on the other hand, retain such data, allowing for retrospective analyses to confirm findings or examine new items of interest (e.g., additional quality criteria) that emerged since the initial assessment. Finally, video has the potential to be analyzed using computer vision techniques. If an accurate computer vision algorithm were developed to assess park attribute quality, the time and cost needed would be substantially reduced, while reliability would improve, given that only one, accurate algorithm is repeatedly used rather than many different human auditors.

There are strengths associated with this study worth mentioning. The sample of parks studied offered a range of features and amenities of varied sizes, shapes, and orientations. This heterogeneity improves generalizability and the relevance of the results to a larger audience. The WVD used in this study provided high-quality videos while being affordable relative to other WVDs. As such, others wishing to use WVD video capture to assess park attributes may not find the cost prohibitive. Another strength was the study design. In-person audits coincided with WVD and UAS video collection. This aspect improves on previous work using a UAS to assess parks where the on-ground and the UAS assessments did not occur simultaneously [33].

This study also has limitations to consider when interpreting the findings. The sample size of parks, although heterogeneous with respect to attributes, was small, as were the samples of different features, amenities, and incivilities. This prohibited the calculation of statistics on agreement in some cases. Therefore, for certain park attributes, the results may not be relevant. In addition, each park was assessed during one daytime period in the summer. While this is consistent with standard procedures, it does not allow for potential sunlight and seasonal effects to be considered in video analyses [27]. This is primarily applicable for examining the UAS. For instance, because foliage obstructions affected UAS visibility, data collection during non-summer months, when trees have fewer mature leaves, could improve UAS capabilities to detect some attribute details. Sorting out such intricate nuances will require extensive additional research.

## 5. Conclusions

Further research is needed to determine how and if a UAS can be used to assess the conditions of park attributes. On the other hand, enhancing or supplanting in-person PARAs with WVD video PARAs are viable options for this purpose. If enhancement is desired, a WVD could be worn while conducting in-person audits. This would provide a permanent account of the park attributes that could be retroanalyzed. Using WVD PARAs only would be less obtrusive and provide a lasting record of the park, but these advantages probably do not outweigh the extra time needed to analyze the videos. However, the current study provides evidence for the eventual utilization of computer vision techniques for automating the video analysis process, which will result in a substantial reduction in the time required.

## Figures and Tables

**Figure 1 ijerph-19-11717-f001:**
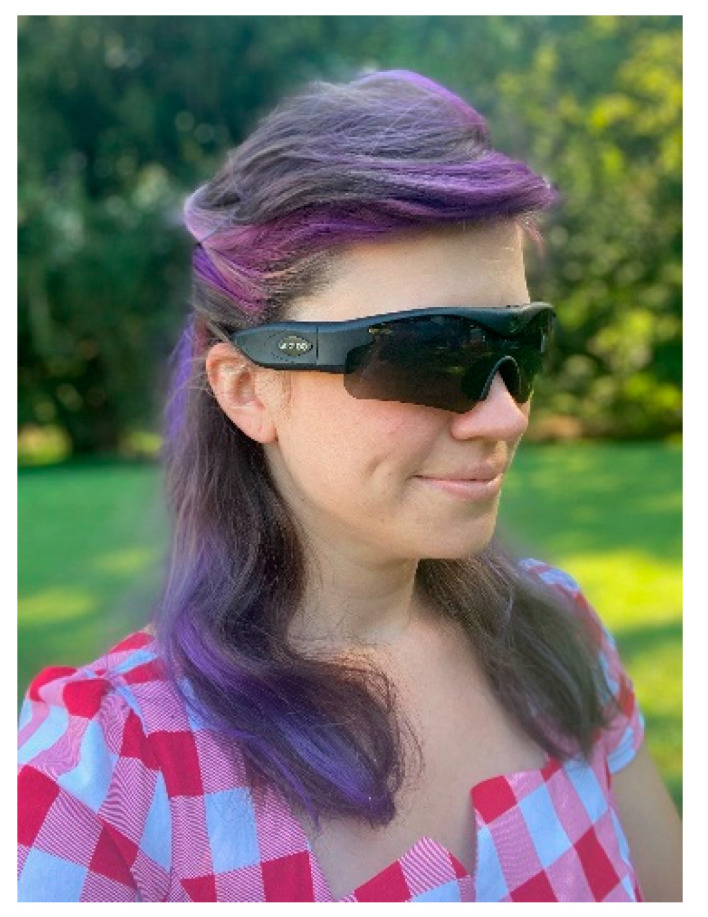
Gogloo E7 SMART wearable video device used in the current study.

**Figure 2 ijerph-19-11717-f002:**
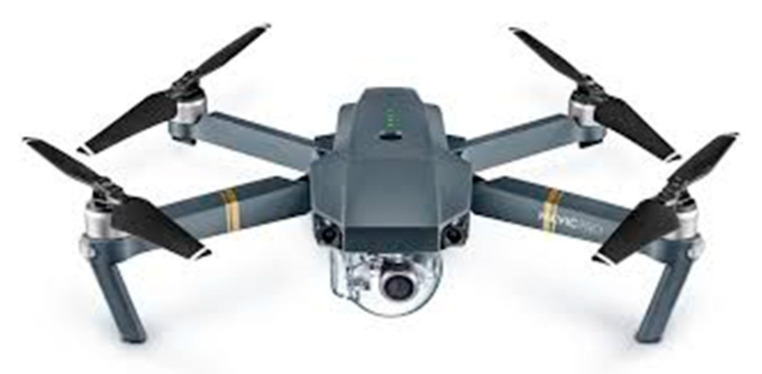
DJI Mavic 2 Pro: the unmanned aerial system used in the current study.

**Table 1 ijerph-19-11717-t001:** Agreement between in-person and WVD and UAS PARAs on the quality of park features and amenities and the severity of incivilities.

	**Kappa (*p* Value, *n*^a^)**	
**Features**	**WVD**	**UAS**
Baseball	0.78 (<0.001; *n* = 14)	0.53 (0.04; *n* = 14)
Basketball Court	0.86 (<0.001; *n* = 22)	0.75 (<0.001; *n* = 12)
Playground	0.87 (<0.001; *n* = 63)	0.38 (0.02: *n* = 37)
Tennis Courts	0.75 (<0.001; *n* = 17)	0.67 (0.01; *n* = 8)
Trails	0.80 (0.005; *n* = 9)	0.33 (0.25; *n* = 4)
**Amenities**	**Kappa (*p* value, *n*^a^)**	
Access points	0.89 (<0.001; *n* = 20)	0.44 (0.20; *n* = 11)
Benches	0.79 (<0.001; *n* = 47)	0.41 (0.12; *n* = 23)
Landscape	0.85 (<0.001; *n* = 25)	0.35 (0.12; *n* = 17)
Picnic tables	0.78 (<0.001 *n* = 16)	0.29 (0.64; *n* = 9)
Trash containers	0.81 (<0.001; *n* = 20)	0.78 (0.02; *n* = 12)
Lighting	0.86 (0.01; *n* = 7)	0.55 (0.03; *n* = 4)
Shelters	0.79 (<0.001; *n* = 13)	0.42 (0.09; *n* = 6)
**Incivilities**	**Kappa (*p* value, *n*^b^)**	
No grass	0.94 (<0.001; *n* = 24)	0.36 (0.04; *n* = 10)
Overgrown grass	0.94 (<0.001; *n* = 27)	0.28 (0.05; *n* = 8)
Broken glass	0.73 (<0.001; *n* = 9)	*
Dog refuse	0.51 (0.006; *n* = 5)	*
Unattended dogs	*	*
Alcohol evidence	0.65 (<0.001; *n* = 10)	*
Evidence of substance abuse	*	*
Graffiti	0.76 (<0.001; *n* = 15)	0.32 (0.02; *n* = 10)
Litter	0.77 (<0.001; *n* = 22)	0.07 (0.73; *n* = 11)
Sex paraphernalia	*	*
Vandalism	0.79 (<0.001; *n* = 6)	*

*n*^a^ = sample sizes of features and amenities rated; *n*^b^ = number of parks where the incivility was present, and thus provided a rating from 1–3; * Kappas could not be calculated due to one or 0 parks displaying those incivilities.

## Data Availability

The data presented in this study are available on request from the corresponding author. The data are not publicly available due to privacy restrictions.

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
