# Peer review of "Assessing Park Quality with a Wearable Video Device and an Unmanned Aerial System"

_ijerph, 2022, doi:10.3390/ijerph191811717_

Round 1

Reviewer 1 Report

As presented the overall merit of the paper is low.

------

5. Conclusions

Further research is needed to determine how and if a UAS can be used to assess the conditions of park attributes.

-------

The authors should complete the research and provide more scientific information and data.

Moreover, it cannot be clear to the reader how the data and information presented in table 1 were collected and calculated.

4. Discussion

The aim of this study was to determined if the quality of park attributes can be accurately ascertained from videos obtained with a WVD and an UAS. Results indicate that WVD videos are viable for assessing the quality of parks features and amenities and the severity of park incivilities. Conversely, the UAS videos were not adequate for assessing most park attributes.

What are the comparison tools? 

It is not clear to the reader how the parameters presented in table 1 was measured and compared.

The presentation of the paper should be improved.

The authors should complete the research and provide more scientific information and data.

Author Response

Thank you for your review

Reviewer 2 Report

This work presents a comparison between two methods to obtain information about parks, aiming to visualize their conditions of use. The paper is well written, the objective and the contribution are well described by the authors. I have some suggestions for the authors.

- Define PARA in abstract and provide more information on text.

- Please, provide a reference to support these sentences:

·         “Video recorders have been embedded into eyewear frames [wearable video device (WVD)] providing point-of-view assessments or attached to aviation devices [unmanned aerial systems (UAS) – drones].”

·         “The Gogloo Engine E7 is a state-of-the-art, point of view WVD, indistinguishable from a pair of normal sunglasses (Figure 1).” The reference of the Gogloo is one the end of the paragraph and it is made of wrong way.

- I suggest the authors to no start any subsection without a text after a section, as in page 3.

- Some subsections are not formatted as in the template, as in sections 2 and 3. Please, correct them.

- The Figure 2 split the text. Correct the paragraph.

- Provide more information on the figure’s legend.

- “(…) is relatively inexpensive (<$3,000)”. If other equipments are not presented, how the authors can affirm that it is inexpensive. Please, review it.

- “Videos were reviews by two authors (…)”. This kind of information should be inserted on the end of the paper and not on the methodology section.

- Format the table as journal’s template.

- “Grubesic and colleagues” use et al.

- The use of only one technique for analysis may not be feasible to be able to provide the results and analysis obtained. Authors should specify other techniques and/or use more statistical techniques.

- Why were processing and machine learning techniques not applied to detect the images obtained?

Author Response

Thank you for your review

Round 2

Reviewer 1 Report

The paper was improved and can be published in the journal.